# Development and Characterization of Polylactide Blends with Improved Toughness by Reactive Extrusion with Lactic Acid Oligomers

**DOI:** 10.3390/polym14091874

**Published:** 2022-05-04

**Authors:** Ramon Tejada-Oliveros, Stefano Fiori, Jaume Gomez-Caturla, Diego Lascano, Nestor Montanes, Luis Quiles-Carrillo, David Garcia-Sanoguera

**Affiliations:** 1Technological Institute of Materials (ITM), Universitat Politècnica de València (UPV), Plaza Ferrándiz y Carbonell 1, 03801 Alcoy, Spain; rateol@epsa.upv.es (R.T.-O.); jaugoca@epsa.upv.es (J.G.-C.); dielas@epsa.upv.es (D.L.); nesmonmu@upvnet.upv.es (N.M.); luiquic1@epsa.upv.es (L.Q.-C.); 2Condensia Quimica S.A., Technical Department, C/Junqueras 16, 11-A, 08003 Barcelona, Spain; s.fiori@condensia.com

**Keywords:** polylactide (PLA), reactive extrusion, improved toughness, environmentally friendly, lactic acid oligomer (OLA), maleinized linseed oil (MLO)

## Abstract

In this work, we report the development and characterization of polylactide (PLA) blends with improved toughness by the addition of 10 wt.% lactic acid oligomers (OLA) and assess the feasibility of reactive extrusion (REX) and injection moulding to obtain high impact resistant injection moulded parts. To improve PLA/OLA interactions, two approaches are carried out. On the one hand, reactive extrusion of PLA/OLA with different dicumyl peroxide (DCP) concentrations is evaluated and, on the other hand, the effect of maleinized linseed oil (MLO) is studied. The effect of DCP and MLO content used in the reactive extrusion process is evaluated in terms of mechanical, thermal, dynamic mechanical, wetting and colour properties, as well as the morphology of the obtained materials. The impact strength of neat PLA (39.3 kJ/m^2^) was slightly improved up to 42.4 kJ/m^2^ with 10 wt.% OLA. Nevertheless, reactive extrusion with 0.3 phr DCP (parts by weight of DCP per 100 parts by weight of PLA–OLA base blend 90:10) led to a noticeable higher impact strength of 51.7 kJ/m^2^, while the reactive extrusion with 6 phr MLO gave an even higher impact strength of 59.5 kJ/m^2^, thus giving evidence of the feasibility of these two approaches to overcome the intrinsic brittleness of PLA. Therefore, despite MLO being able to provide the highest impact strength, reactive extrusion with DCP led to high transparency, which could be an interesting feature in food packaging, for example. In any case, these two approaches represent environmentally friendly strategies to improve PLA toughness.

## 1. Introduction

Currently, there is great concern in terms of environmental pollution, greenhouse gas emissions and depletion of fossil resources due to the intensive use of petroleum-derived and non-compostable or recyclable materials [1,2,3]. This has promoted the development of environmentally friendly and high-performance polymeric materials that are now considered promising alternatives to traditional petroleum-derived polymers [4,5]. Among all these polymers, aliphatic polyesters such as poly(lactide)–(PLA), and bacterial polyesters such as poly(hydroxybutyrate)–(PHB) or poly(hydroxybutyrate-co-valerate)–(PHBV), have gained great attention. These polyesters, together with other biobased polymers such as thermoplastic starch (TPS) or protein-derived polymers, have gained popularity largely due to their easy processing and above all their biodegradation or disintegration in controlled compost soil, thus minimizing environmental impact and carbon footprint. Moreover, PLA offers similar (or even superior) properties compared to most commodities such as poly(styrene)–(PS), poly(ethylene)–(PE) or poly(propylene)–(PP). In addition, PLA can be processed by conventional techniques such as melt spinning [6], injection moulding [7], extrusion [8] or other advanced manufacturing processes such as electrospinning [9], 3D-printing [10], among others.

Polylactic acid (PLA) is one of the most studied aliphatic polyesters and is currently considered the first choice in the emerging market of bioplastics due to its good balance between mechanical, thermal, optical and barrier properties [11]. Moreover, it can undergo biodegradation or disintegration in compost conditions, and it can be synthesized from renewable resources. All these features have led to the increasing use of PLA in the packaging industry [12,13], pharmaceutical applications [14], medical uses [15,16], automotive parts [17] and 3D-printed technology [18]. PLA is obtained through the anaerobic fermentation of sugars derived from starch-rich plants such as corn, sugar cane, beet sugar, potato, among others [19], through direct condensation of lactic acid [20], and by ring opening polymerization of cyclic lactide dimer (ROP) [21].

Despite its relatively good properties, PLA has an intrinsically brittle behaviour and a low elongation at break [22], and this is an important drawback in some technical applications. To overcome this, or at least minimize it, significant research has been carried out with different approaches. One approach is copolymerization with flexible monomers, e.g., poly(lactide)-g-poly(butylene succinate-co-adipate) [23], or long chain aliphatic monomers such as poly(lactic acid-co-ethylene glycol)] [24]. From an industrial point of view, the most commonly used strategy is blending. Binary or ternary blends of PLA with other polymers can reach the desired properties and reduce the final cost of the polymers. With the aim of reducing its high brittleness, PLA has been blended with other aliphatic polyesters that act as impact modifiers, such as poly(3-hydroxybutyrate)–(P3HB) [25], and with other rubbery-like polyesters such as poly(butylene adipate-co-terephthalate)–(PBAT) [26,27], poly(butylene succinate)–(PBS) [28,29] or poly(ε-caprolactone)–(PCL) [30], which provide improved toughness to PLA due to the typical morphology of rubber-like microparticles dispersed in the brittle PLA matrix.

Another feasible option is the use plasticizers. Plasticisers have been widely used in PLA formulations with the purpose of reducing its brittleness and increasing ductile properties, such as elongation at break. It has been observed that plasticization of PLA with triethyl citrate (TEC) [31] results in a noticeable improvement of ductility and elongation at break. Reported, too, has been the synergistic effect of combining two plasticizers such as poly(ethylene glycol)–(PEG) and epoxidized soybean oil (ESBO) [32]. As mentioned above, the environmental issues related to the use of petroleum-derived polymers and additives has led to the assessment of environmentally friendly plasticizers, which can contribute to improve toughness without compromising the overall biodegradability. In this regard, oligomers of lactic acid (OLAs) have been widely proposed as plasticizers for PLA [33], with very interesting results on improved toughness [34,35]. Nevertheless, as with other plasticizers, OLAs lead to a decrease in the glass transition temperature (T_g_) as reported by Burgos et al. [34].

In general, the final properties of plasticized PLA formulations and blends are highly dependent on their miscibility/compatibility. In some cases, partial or total immiscibility are obtained, which leads to lowering the plasticizer or blend performance [36,37]. This immiscibility usually leads to phase separation with a brittle PLA matrix in which spherical microparticles of a flexible polymer or excess plasticizer can be observed. To overcome this lack of miscibility, compatibilizers are often used in polymer blends [38,39]. The incorporation of a compatibilizer agent can be carried out by two main processes, ex situ (non-reactive) compatibilization or in situ (reactive) compatibilization [40]. The first method is based on the use of a copolymer with different comonomers which are able to interact with both components in a PLA blend or plasticized PLA formulation. Typically, these tailored copolymers provide better interfacial adhesion and consequently improve overall mechanical performance [41]. In the case of OLAs, they are characterized by high miscibility/compatibility with PLA since they offer the same polyester structure [42]. The second compatibilization method is by reactive extrusion (REX) which is characterized by triggering some chemical reactions between PLA chains and additives, during the extrusion process [43]. It is important to bear in mind that PLA end chains contain hydroxyl or carboxyl groups, and this can readily react with a wide series of additives [44]. Usually, REX is triggered by using free radical initiators which promote reaction during the extrusion process. Among others, dicumyl peroxide (DCP) has proven to be an effective initiator for REX with a wide variety of polymers and additives [45]. DCP has been used to increase compatibility in binary polymer blends with excellent results. As Mehmood et al. [46] have reported, a remarkable improvement on the compatibility between PLA and Arabic gum is obtained by reactive extrusion with the DCP initiator. Together with OLAs, epoxidized vegetable oils (EVOs) have gained interest since they represent an environmentally friendly technical solution to overcome the intrinsic brittleness of PLA. It is worth noting the increasing use of epoxidized soybean oil (ESBO) in plasticized PLA formulations [47] and more recently, maleinized linseed oil (MLO) has been successfully used to improve PLA toughness [48]. The use of vegetable oil derivatives is a sustainable solution in PLA formulations [49,50]. Due to its high reactive maleic anhydride (MA) groups, MLO can provide some different effects on PLA and its blends such as plasticization, branching, chain extension, crosslinking and compatibilization [49]. MLO has also provided good compatibilization in other aliphatic polyester formulations, such as poly(butylene succinate)–(PBS) filled with almond shell flour [51], with a remarkable increase in ductile and resistant properties of composites.

The main aim of this work is to improve the low intrinsic toughness of PLA by using reactive extrusion (REX) with dicumyl peroxide (DCP). The novelty of this lies in the use of different environmentally friendly plasticizers, namely, oligomers of lactic acid (OLAs) and maleinized linseed oil (MLO), in order to obtain largely bio-based materials that can replace conventional fully petroleum-derived polymers The effect of the REX process on mechanical properties, morphology, thermal and thermomechanical behaviour is described in this work, as well as the visual appearance and the wetting properties.

## 2. Materials and Methods

### 2.1. Materials

The base polymer was a poly(lactide)–PLA with a commercial grade PURAPOL L130 supplied by Total Corbion PLA (Amsterdam, the Netherlands). This PLA grade contains 99% L-isomer, a density of 1.24 g/cm^3^ and a melt flow index (MFI) of 16 g/10 min at 210 °C. Oligomers of lactic acid (OLA) were supplied by Condensia Química S.A. (Barcelona, Spain), under the tradename Glyplast OLA2. As shown in its technical data sheet, this OLA has a viscosity of 90 mPa s at 40 °C, a density of 1.10 g/cm^3^, an ester content >99%, a maximum acid index of 2.5 mg KOH/g and a maximum water content of 0.1%. Reactive extrusion was carried out with two different strategies: one consisting of the use of a free radical initiator, and a second one by using a maleic anhydride functionalized vegetable oil. Dicumyl peroxide was used as initiator for reactive extrusion. This was supplied by Sigma Aldrich (Madrid, Spain) and has a purity of 98%. The maleinized linseed oil (MLO) was a commercial-grade VEOMER LIN, supplied by Vandeputte (Mouscron, Belgium). This maleinized oil has a viscosity of 10 dPa s at 20 °C and a maximum acid index comprised in the 105–130 mg KOH/g. Figure 1 depicts the chemical structure of all used materials in this work.

### 2.2. Preparation of PLA/OLA Blends

With the aim of removing residual moisture, PLA was dried at 65 °C for 48 h in a dehumidifier dryer model TCN 115 from Labprocess Distribuciones S.L. (Barcelona, Spain). After this, the appropriate amount of OLA (10 wt.%), DCP and MLO (see Table 1 for compositions and coding) were added to PLA and fed into the hopper of the extruder.

All the formulations in Table 1 were processed by a twin screw co-rotating extruder with a diameter of 25 mm and a length to diameter (L/D) ratio of 24. The formulations and parameters have been selected according to the following previous works [52,53]. This extruder was provided by Construcciones Mecánicas Dupra S.L. (Alicante, Spain). The temperature profile used was 160 °C–175 °C–185 °C and 190 °C from the hopper to the die. The obtained strands were cooled down to room temperature and cut into pellets for further processing by injection moulding in a Meteor 270/75 from Mateu & Solé (Barcelona, Spain). The temperature profile for the injection moulding process was 175 °C (hopper)–180–185 °C and 190 °C (injection nozzle). The filling and cooling times were set to 1 and 10 s, respectively, and the applied clamp force was 75 ton.

### 2.3. Characterization of PLA/OLA Blends

#### 2.3.1. Mechanical Properties

Mechanical characterization of neat PLA and PLA/OLA blends were obtained by tensile, Charpy and Shore D hardness tests. Tensile tests were carried out in a universal testing machine model ELIB 50 from S.A.E. Ibertest (Madrid, Spain) on dog-bone specimens according to ISO 527-1:2012. The crosshead speed rate was set to 10 mm/min and a load cell of 5 kN was used. The impact strength was obtained from Charpy tests using a 6-J pendulum from Metrotec S.A. (San Sebastián, Spain), on unnotched samples with dimensions 80 × 10 × 4 mm^3^ as indicated by ISO 179-1:2010. With regard to hardness, a Shore durometer mod 76-D from J. Bot Instruments (Barcelona, Spain) was used. The Shore-D hardness values were obtained after 15 s to obtain reliable values as recommended by the above-mentioned standard. All mechanical tests were carried out in, at least, 6 specimens and the average values of the main parameters were calculated.

#### 2.3.2. Morphology Characterization

The morphology of the PLA/OLA system subjected to different REX processes was obtained from fractures’ surfaces from impact test specimens with a field emission scanning electron microscope (FESEM) ZEISS ULTRA 55 from Oxford Instruments (Abingdon, UK). To provide electric conductivity to the polymeric samples, a sputtering process was carried out in a EMITECH sputter-coater model SC7620 from Quorum Technologies, Ltd. (East Sussex, UK). The working distance was set to 4 mm and the acceleration voltage was 2 kV.

#### 2.3.3. Thermal Analysis

The most important thermal properties of neat PLA and PLA/OLA blends subjected to REX with different additives were obtained by differential scanning calorimetry (DSC). A Mettler –Toledo calorimeter model 821 (Schwerzenbach, Switzerland) was used to collect the thermograms. To enhance reliable results, a sample weight of 5 to 7 mg was placed into sealed standard aluminium crucibles with a volume of 40 μL. Samples were subjected to a dynamic thermal program with three stages. A first heating ramp from 25 °C to 200 °C at 10 °C/min was applied to remove the thermal history related to processing conditions. Then, a controlled cooling was scheduled from 200 °C to −30 °C at −10 °C/min. Finally, a second heating step was programmed from 30 °C up to 300 °C. All DSC tests were done in triplicate in a nitrogen atmosphere with a flow rate of 66 mL/min. The degree of crystallinity ($\chi_{c}$%) was calculated following Equation (1):(1)$$\chi_{c}\left(\%\right)=\left(\frac{∆H_{m}-∆H_{cc}}{∆H_{m}^{0}}\right)\times \frac{100}{w}$$
where $H_{m}{\;\mathrm{and}\;H}_{cc}$ stand for the melt and cold crystallization enthalpies, respectively. $H_{m}^{0}$ represents the theoretical melt entalpy for a 100% crystalline PLA and was taken as 93.7 J/g as reported in the literature [54,55]. In this equation, w represents the weight percentage of PLA.

Thermal degradation of neat PLA and PLA/OLA blends subjected to REX with different additives were studied by thermogravimetry (TGA) in a TGA thermobalance model 1000 from LINSEIS (Selb, Germany). Samples with an average weight of 15–20 mg were placed into alumna crucibles with a volume of 70 μL. A temperature ramp from 30 °C to 700 °C was programmed at a heating rate of 20 °C/min in nitrogen atmosphere. All TGA tests were run in triplicate.

#### 2.3.4. Thermomechanical Characterization

Dynamic mechanical thermal characterization (DMTA) of neat PLA and PLA/OLA blends were obtained in a DMTA analyser from Mettler–Toledo, model DMA1 (Schwerzenbach, Switzerland), working in single cantilever flexural conditions. Samples with dimensions 20 × 6 × 2.7 mm^3^ were subjected to a dynamic sweep from 30 °C to 140 °C at a heating rate of 2 °C/min. The selected frequency was 1 Hz and the maximum cantilever deflection was set to 10 µm. DMTA tests were run in triplicate and averaged.

#### 2.3.5. Colour and Wetting Characterization

A Konica CM-3600d Colorflex-DIFF2 spectrophotometer from Hunter Associates Laboratory, Inc. (Reston, VA, USA.) was used for colour measurements. L*a*b* colour coordinates were measured with L* representing the luminance, a* the colour coordinate from green (a* < 0) to red (a* > 0) and b* standing for the colour coordinate from blue (b* < 0) to yellow (b* > 0). The yellowing index was calculated as recommended by ASTM E313. At least 10 different measurements were done on flat specimens and the average colour coordinates were obtained.

The surface wetting properties were obtained using an optical goniometer EasyDrop Standard model FM140 from KRÜSS GmbH (Hamburg, Germany) that is equipped with a video capture accessory kit. Double distilled water was used for contact angle measurements using the Drop Shape Analysis SW21; DSA1 software. Flat specimens with dimensions 80 × 10 × 4 mm^3^ were used to obtain the water contact angle (θ_w_) at room temperature. At least 10 different measurements were done and the obtained θ_w_ were averaged.

## 3. Results

### 3.1. Effect of REX on Mechanical Properties of PLA/OLA Blends

Table 2 gathers the main properties of PLA/OLA formulations subjected to REX, obtained from mechanical tests. Neat PLA is a brittle polymer and this is reflected in its mechanical properties, with a tensile modulus E of 2912 MPa, a maximum tensile strength (σ_max_) of 47.0 MPa and a remarkable low elongation at break (ε_b_) of 7.1% which is responsible for low toughness [56]. Addition of 10 wt.% OLA led to a slight increase in tensile modulus up to 3138 MPa but, as expected, the maximum tensile strength decreases to 30.8 MPa due to the plasticization effect. Nevertheless, the elongation at break was reduced by 38% with regard to neat PLA. Other works have reported a clear plasticization effect provided by OLA on PLA with a decrease in both tensile strength and modulus and a noticeable improvement on elongation at break [34,57]; but it has also been reported that some plasticizers, despite not showing the expected plasticization effect on mechanical properties, do provide a decrease in the corresponding glass transition temperature (T_g_) [52,58].

It is worth noting the effects of the reactive extrusion (REX) with dicumyl peroxide (DCP) and maleinized linseed oil (MLO). REX with DCP leads to interesting effects on the base PLA/OLA blend. On one hand, the tensile modulus remains almost constant with values close to 3000 MPa while, as expected, the tensile strength is increased up to 35.7 MPa and 36.8 MPa for a DCP content of 0.1 phr and 0.3 phr respectively. DCP deposition promotes free radical formation, and subsequently, hydrogen abstraction from both PLA polymer chains and OLA molecules can occur, which could provide an increase in the interaction between these two components (grafting OLA molecules on PLA chains). On the other hand, the increased interaction between the components results in the formation of some cross-linked structure, which causes a restriction in the mobility of the polymer chains, impeding the ability of the blends to dissipate energy under tensile load, with a subsequent decrease in the elongation at break with respect to neat PLA. A scheme of the plausible reaction mechanism during REX of PLA/OLA blend with DCP can be observed in Figure 2. Accordingly to these results, Monika et al. [59] reported similar effects in ternary blends composed of poly(lactide)—PLA, poly(butylene succinate)—PBS and chitosan. REX with DCP leads to an increase in both modulus and tensile strength but the elongation at break of the ternary blend was lower than that of neat PLA.

The reactive extrusion with maleinized linseed oil (MLO) is completely different. It has been reported that reactive extrusion of aliphatic polyesters and blends with MLO can lead to several overlapped processes such as chain extension, branching, compatibilization and plasticization [60,61,62]. At low concentration, MLO does not provide important changes in tensile properties (E, σ_max_ and ε_b_), compared to an uncompatibilized base PLA/OLA blend, thus suggesting compatibilization is not achieved and plasticization effects are very restricted. Sarasini et al. [63] observed similar results in PHBV/PBAT blends with coffee silverskin compatibilized with MLO. Nevertheless, by increasing the MLO content up to 6 phr, an important effect can be depicted. On the one hand, the tensile strength increases up to 41.7 MPa (which represents a % increase of 35%) and, on the other hand, an interesting increase in elongation at break can be detected with values of 8.1% (which represents an increase of 84% with regard to the uncompatibilized base PLA/OLA blend). It has been reported that MLO, at these concentrations, can exert some plasticization due to lubrication of polymer chains which lead to increased mobility with a subsequent improvement of ductility. Moreover, due to its particular structure (attached maleic anhydride groups), MLO can react with terminal–OH groups present in both PLA chains and OLA oligomers, which in turn provides a slight compatibilization effect with a subsequent increase in tensile strength [64,65].

Table 2 also gathers the main results regarding impact strength of PLA/OLA blends subjected to REX with different additives. Neat PLA is a brittle polymer with a very low toughness. Its impact strength is 39.3 kJ/m^2^ [65]. The addition of only 20 wt.% OLA does not provide a noticeable improvement of toughness, but the average impact strength is slightly higher (42.4 kJ/m^2^). Nevertheless, REX with DCP has a remarkable positive effect on toughness with impact strength values ranging from 44.5 kJ/m^2^ up to 51.7 kJ/m^2^ for 0.1 phr and 0.3 phr DCP, respectively. Accordingly, to other tensile properties, the impact strength of PLA/OLA blends is remarkably improved by REX with 0.3 phr DCP. Higher DCP loadings lead to more free radical formation during REX, and this allows anchoring of OLA molecules into PLA polymer chains. This has a positive effect on PLA–OLA interaction and, subsequently, the impact strength is improved, thus confirming that REX with DCP is an interesting approach to improve the low intrinsic toughness of PLA and its blends [66]. Similar effects can be observed by REX with MLO with impact strength values of 52.3 kJ/m^2^ and 59.5 kJ/m^2^ for 3 phr and 6 phr MLO, respectively. The impact strength obtained with 6 phr MLO is 51.4% higher compared to neat PLA. Quiles et al. [67], reported that the multifunctional modified vegetable oil could act simultaneously as plasticizer and compatibilizer. Finally, with regard to Shore D, no remarkable changes in Shore D hardness values can be observed in all developed materials, with values ranging between 78 and 82.

### 3.2. Effect of REX on Morphology of PLA/OLA Blends

Figure 3 gathers the fracture surface morphology of impact specimens observed by field emission scanning electron microscopy (FESEM) of neat PLA, and PLA/OLA blends subjected to REX with different strategies. Figure 3a corresponds to the fracture surface of neat PLA with the typical brittle fracture surface characterized by a smooth surface and the presence of different microcracks. This smooth surface is representative for the low plastic deformation observed in neat PLA, which is in accordance with the brittle behaviour mentioned above [68]. Figure 3b depicts the fracture image corresponding to uncompatibilized PLA/OLA blend which is noticeably different from that of neat PLA. In particular, the smooth surface characteristic of a brittle behaviour has changed to a rougher fracture surface with both micro- and macrocrack formation. Lascano et al. [52] observed this morphology on PLA-based formulations with different OLA content. On the other hand, some spherical shapes can be detected (see white arrows), which can be attributed to partial miscibility between PLA and OLA. This leads to a stress concentration phenomenon and, subsequently, the impact strength is not improved, as mentioned above. This morphology, with a rough surface and dispersed spherical voids, is characteristic of phase separation due to restricted miscibility between PLA and OLA which, in turn, has a negative effect on mechanical properties as described previously [69,70]. Figure 3c,d show the morphology of the fractured surfaces corresponding to PLA/OLA blends subjected to REX with 0.1 and 0.3 phr DCP, respectively. REX with 0.1 DCP (Figure 3c) shows a surface morphology without spherical voids which indicates that REX with DCP positively contributes to obtain a more homogeneous matrix which, in turn, has a positive effect on overall mechanical properties. REX with 0.3 DCP (Figure 3d) shows a rougher surface and almost inexistent spherical voids. This suggests increased compatibility between PLA and OLA through free radical formation which allows chemical anchoring of OLA molecules into the PLA polymer chains. This compatibilization effect is reflected by an increase in toughness, as observed before. Similar results were reported by Akos et al. [71], in PLA/PCL blends subjected to reactive compatibilization with DCP. Figure 3e shows the fracture morphology of a PLA/OLA blend subjected to REX with 3 phr MLO. As can be seen, the spherical voids appear again but, in this case, these voids are related to the modified vegetable oil, with restricted miscibility in the PLA/OLA matrix. Despite this, some filaments can be detected, which are directly related to improved ductile behaviour. This situation is more pronounced in PLA/OLA blend subjected to REX with 6 phr MLO (Figure 3f) with smaller spherical domains and the presence of more filaments, which are responsible for improved ductility [49].

### 3.3. Effect of REX on Chemical Properties

The chemical composition of injection-moulded samples from each blend was evaluated by means of Fourier transformed infrared spectroscopy (FTIR). Figure 4 shows the FTIR spectrums of all the blends developed in this work from 4000 to 600 cm^−1^. Regarding neat PLA, there are several identifiable bands. Two of the main bands are located at 1750 cm^−1^and at 1080 cm^−1^, which are related to the carbonyl C-O-C stretching bonds [72]. A low intensity peak at 1450 cm^−1^ is observed, which is ascribed to the C-H stretching in methyl groups. Another characteristic peak is located at 1043 cm^−1^, which is related to the C-CH3 stretching vibration. These bands appear in all the spectra recorded due to PLA being the base of all the blends. When incorporating OLA to PLA, the spectra do not suffer significant changes. The most relevant observation is an increase in the intensity of the peak/band at 1080–1100 cm^−1^, which is related to the -CO- functionality in the oligomer and the polymer [73], which could mean certain chemical interaction between them. DCP thermically decomposes during reactive extrusion, so it does not present noticeable changes in the FTIR spectra of the samples that contain it [74]. Finally, with regard to the presence of MLO in the blend, a slight increase in the intensity of the peak located at 1158 cm^−1^ can be appreciated, which is indicative of the C-O-C, C-O and C-C stretching vibrations in ester groups [75]. This could be a sign of linkage between maleic anhydride functionalities in MLO with polylactide and OLA. These results suggest that there is a positive interaction between PLA and OLA chains, and that MLO has been positively inserted in the blends.

### 3.4. Effect of REX on Thermal Properties of PLA/OLA Blends

Figure 5 shows the DSC thermograms corresponding to the second heating cycle after removing the thermal history related to processing. On the other hand, Table 3 gathers the main thermal parameters obtained from DSC runs (2nd heating cycle). With regard to neat PLA, the glass transition temperature (T_g_) and the melt peak temperature (T_m_) are located at 63.3 °C, and 173.4 °C, respectively. The cold crystallization is not clearly observed but a very wide peak can be detected in the 120–150 °C range [76]. This PLA is characterized by a degree of crystallinity (χ_c_%) of 20%. OLA addition leads to a noticeable decrease in T_g_ down to values of 49.8 °C which suggests a plasticization effect, but this is not reflected in the overall mechanical properties as previously described. Despite OLA being able to provide a noticeable decrease in T_g_, some recent works have reported a limited effect on mechanical properties of OLA-plasticized PLA formulations [34,57]. On the other hand, the cold crystallization process is clearly identified as an exothermic process with a peak temperature of 97.3 °C. This effect is typical of a plasticizer since the lubrication effect OLA provides to PLA chains promotes increased chain mobility which, in turn, leads to lowering the characteristic cold crystallization temperatures of neat PLA [77]. In this case, the degree of crystallinity is slightly reduced to 16.2% when OLA is added to PLA. Other research works have reported an increase in crystallinity with OLA addition. Lascano et al. [52] reported an increase in χ_c_% of PLA with OLA, which was attributed to a good miscibility between OLA and the PLA grade used. As can be seen in Figure 5, REX has important effects on thermal properties of PLA/OLA blends. REX with DCP does not lead to a noticeable change in T_g_ with values of 50.5 °C and 49.8 °C for 0.1 phr DCP and 0.3 phr DCP, respectively. An important change is observed in the cold crystallization peak which is wider if compared to the base PLA/OLA blend. The most relevant change is a decrease in χ_c_% down to values of 7.8% and 2.4% for REX with 0.1 phr DCP and 0.3 phr DCP, respectively. This is directly related to a disruption of the crystal structure promoted by REX with DCP, since OLA molecules are attached to PLA polymer chains, and these branched chains cannot rearrange to a packed crystal structure. Yang et al. [78] reported similar results in PLA formulations subjected to REX with DCP with triallyl isocyanurate (TAIC). With regard to REX with MLO, two different effects can be observed.

At low MLO concentration (3 phr) the thermal properties remain almost invariable compared to the base PLA/OLA blend with a slight increase in T_g_ due to anchorage of maleic anhydride into PLA polymer chains and OLA molecules (mainly through the reaction with –OH groups in terminal position). Nevertheless, addition of 6 phr MLO during REX leads to a more noticeable increase in T_g_ up to 56.5 °C due to the aforementioned reaction of MLO with both PLA and OLA. MLO also disrupts the crystal structure and this is evidenced by two different phenomena. On the one hand, the χ_c_% is remarkably reduced to 6.7% and, on the other hand, the cold crystallization process is shifted to higher temperatures, with a peak temperature T_cc_ of 105.4 °C. As has been reported by Ferri et al. [49], the maleic anhydride groups attached to the carbon–carbon double bonds of linseed oil readily react with –OH groups in both PLA and OLA, leading to a combination of overlapped phenomena, including plasticization, chain extension, crosslinking, branching and compatibilization, with a clear disruption of the crystal structure. With respect to the melt peak temperature, T_m_, it remains almost invariable with values ranging from 171 to 173 °C. Avolio et al. [57] reported similar thermal behaviour in PLA blends with two selectively functionalized oligomers of lactic acid, a carboxyl (OLA-COOH) and an hydroxyl (OLA-OH) end-capped.

The thermal degradation of PLA/OLA blends subjected to REX was studied by thermogravimetry (TGA). Figure 6a shows the TGA thermograms, while the first derivative (DTG) profiles are gathered in Figure 6b. Neat PLA degrades in a single step process with characteristic degradation temperatures of T_5%_ (temperature for a 5 wt.% mass loss) and T_deg_ (temperature corresponding to the maximum degradation rate) of 312 °C and 359 °C, respectively (see Table 4). These values are in total agreement with those reported to PLA [79]. As can be seen in Figure 6a, the thermal stability of the PLA/OLA blends is reduced due to the chain scission phenomenon caused by the lower molecular weight of OLA. The onset degradation temperature is reduced to 287.6 °C as reported by Lascano et al. [52] in PLA/OLA blends with improved toughness and shape memory behaviour. The effect of REX with DCP and MLO shows, rather, differences. Since DCP decomposition promotes free radical formation, it contributes to reducing the thermal stability of the PLA/OLA blend. Therefore, the onset degradation temperature (T_5%_) decreases down to 282.3 °C and 275 °C with 0.1 and 0.3 phr DCP, respectively. This effect has also been reported by Rytlewski et al. [80] in PLA formulations containing different amounts of DCP. In contrast, PLA/OLA blends subjected to REX with MLO provide a noticeable improvement of the thermal stability with T_5%_ values of 312.3 °C and 316.7 °C for MLO contents of 3 phr and 6 phr, respectively. This phenomenon could be related to the reaction of maleic anhydride groups with end-capped hydroxyl groups in both PLA and OLA which promotes different phenomena, including chain extension, branching and crosslinking, which have a positive effect on overall thermal stability [81]. With regard to the maximum degradation rate temperature (T_deg_), it remains almost constant in the 352–358 °C range.

### 3.5. Dynamic Mechanical Behaviour of PLA/OLA Blends

Dynamic mechanical properties as a function of temperature were obtained by DMTA characterization. Figure 7a shows the evolution of the storage modulus (G’), whereas Figure 7b offers the dynamic damping factor (tan δ) with increasing temperature. In neat PLA, the α-relaxation, related to its glass transition, can be identified by a two-or-three-fold decrease in G’. The T_g_, calculated following the tan δ peak criterion, is located at 71.8 °C (see Table 5). Below its T_g_, PLA has a stiff and brittle behaviour with a G’ value of 1251 MPa at 40 °C. Above its T_g_, G’ has been remarkably reduced, thus leading to a rubbery-like behaviour with a G’ value of 1.6 MPa at 80 °C. The cold crystallization process can be detected by an increase in G’ in the 90–95 °C range, since rearrangement to a packed structure leads to an increase in stiffness [82,83]; after the cold crystallization, G’ increases up to 43.8 MPa. The base PLA/OLA blend shows some interesting changes in G’. It is worth noting a decrease in T_g_ down to 65.5 °C as observed previously by DSC characterization which gives evidence of the plasticization effects OLA can provide to PLA, despite this not being reflected at a macroscopic level with a relatively low elongation at break. Moreover, the characteristic G’ below T_g_ is lower than neat PLA, with a value of 1076 MPa, which suggests a clear plasticization effect as reported by Noivoil et al. [84] in PLA blends with thermoplastic starch, compatibilized with OLA-grafted starch. REX with 0.1 phr DCP leads to T_g_ and G’ values similar to neat PLA whereas REX with higher DCP content (0.3 phr) leads to similar values to the base PLA/OLA blend. This could be related to the stronger compatibilization phenomena provided by 0.3 phr DCP compared to 0.1 phr DCP. Ma et al. [85] observed similar effects on PLA/PBAT blends compatibilized through REX with DCP. REX with MLO leads to a slight increase in T_g_ with values around 68 °C. It is worth noting the G’ value for the PLA/OLA blend subjected to REX, 1235 MPa, almost identical to neat PLA but, as indicated previously, this blend possesses a remarkably improved toughness. Another interesting phenomenon is the cold crystallization shift. REX has a clear effect on decreasing the characteristic temperatures of this process, in agreement with the results obtained by DSC [86].

### 3.6. Colour Measurement of PLA Blends

The visual appearance of PLA blends with OLA is important, especially in application for packaging. Due to its semicrystalline nature, neat PLA is translucent (see Figure 8) with some transparency [87]. This is related to different refractive indexes of the crystalline and the amorphous phases [88]. The base PLA/OLA blend is also translucent, but a noticeable decrease in transparency is observed. REX has a direct effect on visual appearance. REX with DCP provides PLA/OLA blend with high transparency due to the disruption of the crystal structure which, in turn, is responsible for a decrease in crystallinity, as observed by DSC. In the case of PLA/OLA blends subjected to REX with MLO, samples are translucent with a slight yellow colour. Despite MLO also providing an important disruption of the crystal structure, transparency is not reached since MLO is not fully miscible with the base PLA/OLA blend.

In addition to the previous qualitative assessment, the CIE L*a*b* colour coordinates have been measured for each material (see Table 6). L* stands for the luminance and this colour coordinate changes in a narrow range comprising between 40 and 46. Nevertheless, OLA addition leads to some yellowing as observed by Burgos et al. [81]. This yellowing is much more intense in PLA/OLA blends subjected to REX with MLO due to the intrinsic yellow colour of linseed oil and its maleinized derivative. With regard to the a* coordinate, this reflects the colour change between the green (a* < 0) and red (a* > 0). All samples are characterized by a very low a* coordinate, close to 0, since all materials are translucent or transparent. Despite this, it seems that MLO leads to slightly lower a* values, as observed by Quiles-Carrillo et al. [89], in PA1010/PLA blends compatibilized with MLO. Regarding the b* coordinate, this represents the colour change between the blue (b* < 0) and yellow (b* > 0). All the PLA/OLA blends are characterized by positive b* values, which is representative for some yellowing. As has been qualitatively observed in Figure 8, PLA/OLA blends subjected to REX with MLO, offer the highest b* values of all developed materials. This is directly related to the intrinsic yellow colour of MLO [90]. Table 6 also contain the yellow index (YI) of all PLA-based materials. As expected, the YI increases with OLA addition, and this increase is much pronounced in PLA/OLA blends subjected to REX with MLO.

### 3.7. Wetting Properties of PLA/OLA Blends

In addition to the visual appearance, the surface wetting properties have also been evaluated. The water contact angle (θ_w_) was measured on PLA and PLA/OLA blends. High θ_w_ values are representative for low affinity to water. As can be seen in Figure 9, all samples have a water contact angle, θ_w_ >65°, which could be considered as the hydrophobic behaviour threshold [91]. Neat PLA shows a θ_w_ of 88.2° which stands for a typical hydrophobic polymer. OLA addition leads to a slight decrease in θ_w_ down to 83.9° since oligomers of lactic acid have lower molecular weight and are more hydrophilic, resulting in a slightly reduced hydrophobicity as observed by Darie-Niţă et al. [35]. In relation to the PLA/OLA blend subjected to REX with MLO (6 phr), the water contact angle is similar to that of neat PLA, with a value around 87.5°, which could be related to the change in crystallinity [92], and the intrinsic hydrophobic nature of MLO, as suggested by Carbonell-Verdu et al. [69].

## 4. Conclusions

This work addresses the development of PLA-based formulations with improved toughness by blending with oligomers of lactic acid (OLA). Despite the blend containing 20 wt.% OLA leading to a noticeable decrease in the glass transition temperature from 63.3 °C (neat PLA) down to values of 49.8 °C, the effects on toughness are not clearly observed. Neat PLA is characterized by an impact strength of 39.3 kJ/m^2^ and this is slightly improved in the PLA/OLA blend containing 20 wt.% OLA. In order to improve the impact strength, reactive extrusion (REX) with different additives was studied, namely, 0.1 and 0.3 phr of dicumyl peroxide (DCP), and 3.6 phr of maleinized linseed oil (MLO). REX with DCP leads to a noticeable disruption of the crystal structure in the PLA/OLA blend. Decomposition of DCP into free radicals during REX allows anchoring of OLA molecules into PLA polymer chains. It is worth noting a remarkable increase in toughness in PLA/OLA blend subjected to REX with 0.3 DCP, with an impact strength of 51.7 kJ/m^2^, thus showing the efficiency of this strategy. The second approach is REX with MLO, since maleic anhydride groups in MLO can readily react with end-capped –OH groups contained in both PLA and OLA. MLO also provides a disruption of the crystal structure of PLA which is more pronounced for 6 phr MLO. Nevertheless, MLO provides the PLA/OLA blend with interesting ductile properties and, what is more important, with a high impact strength value of 59.5 kJ/m^2^. Another interesting finding is transparency. REX with DCP leads to high-transparency materials due to the dramatic decrease in crystallinity as a consequence of the disruption of the crystal structure. With regard to the use of MLO, the obtained materials are translucent, with a slight yellow colour as a consequence of the intrinsic yellow colour of MLO. Therefore, this work provides two innovative strategies to obtain high toughness PLA/OLA formulations by reactive extrusion (REX). Depending on the additive used for the REX process, it is possible to tailor transparency, impact strength, as well as other mechanical and thermal properties.

## Figures and Tables

**Figure 1 polymers-14-01874-f001:**
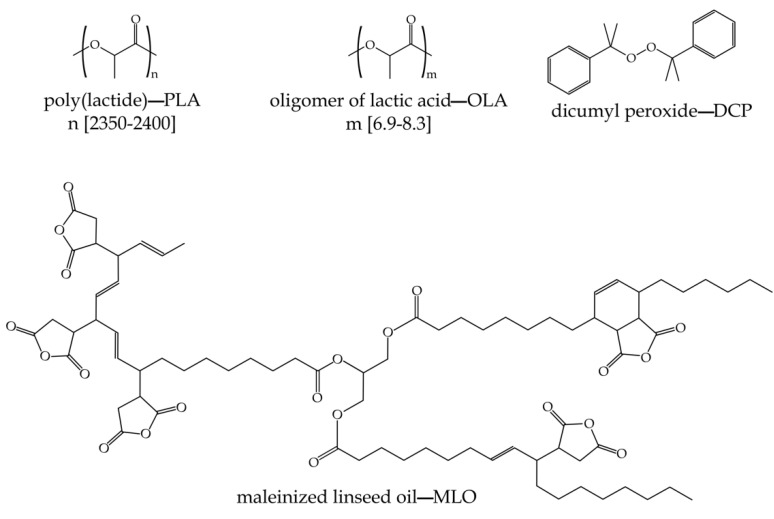
Schematic representation of the blend components: poly(lactide)—PLA, and oligomer of lactic acid—OLA, and additives for reactive extrusion (REX), dicumyl peroxide—DCP, and maleinized linseed oil—MLO.

**Figure 2 polymers-14-01874-f002:**
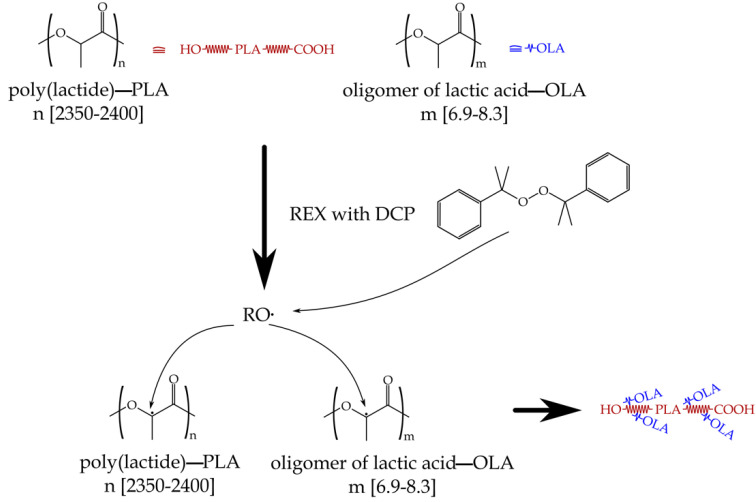
Schematic of the plausible reaction between PLA and OLA during reactive extrusion (REX) with dicumyl peroxide (DCP).

**Figure 3 polymers-14-01874-f003:**
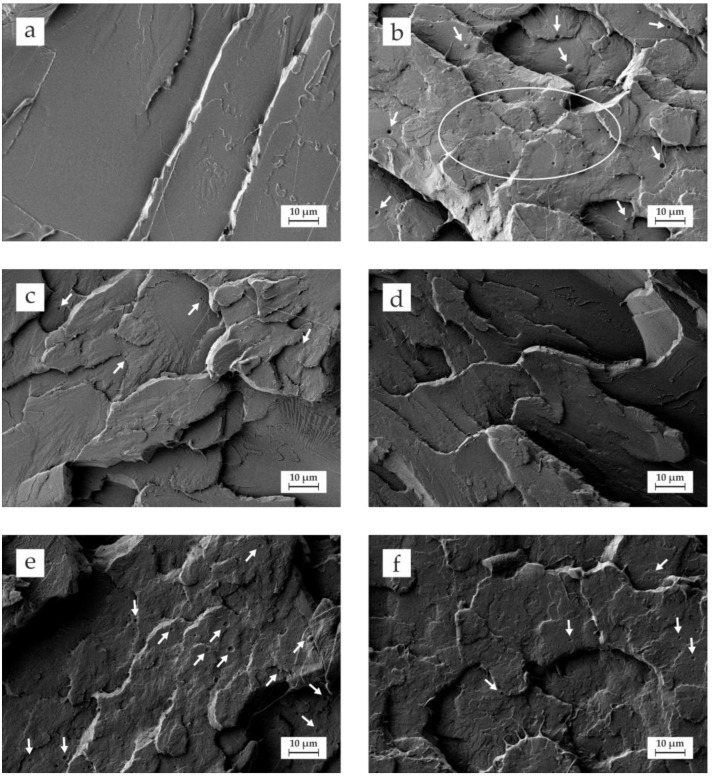
Field emission scanning electron microscopy (FESEM) images at 1000× of the fractured surfaces of (**a**) neat PLA; (**b**) PLA/OLA; (**c**) PLA/OLA/0.1DCP; (**d**) PLA/OLA/0.3DCP; (**e**) PLA/OLA/3MLO; (**f**) PLA/OLA/6MLO.

**Figure 4 polymers-14-01874-f004:**
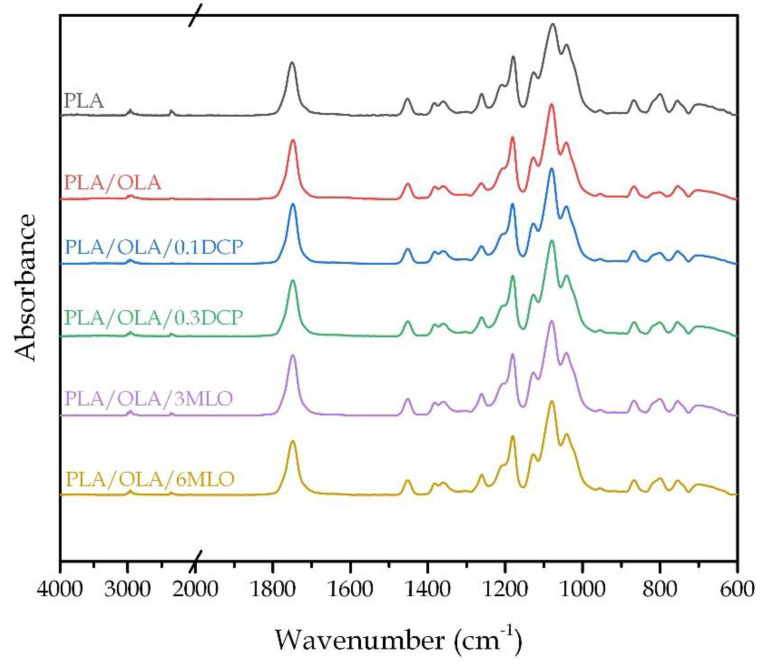
Fourier Transformed Infrared Spectroscopy (FTIR) of all the PLA/OLA blends in the wavenumber range from 4000 to 500 cm^−1^.

**Figure 5 polymers-14-01874-f005:**
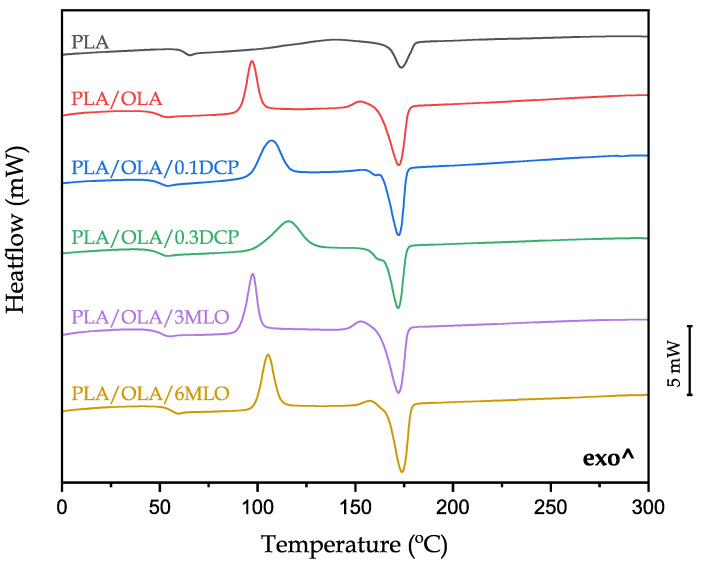
Differential scanning calorimetry (DSC) thermograms of neat PLA, base PLA/OLA blend and PLA/OLA blends subjected to reactive extrusion (REX).

**Figure 6 polymers-14-01874-f006:**
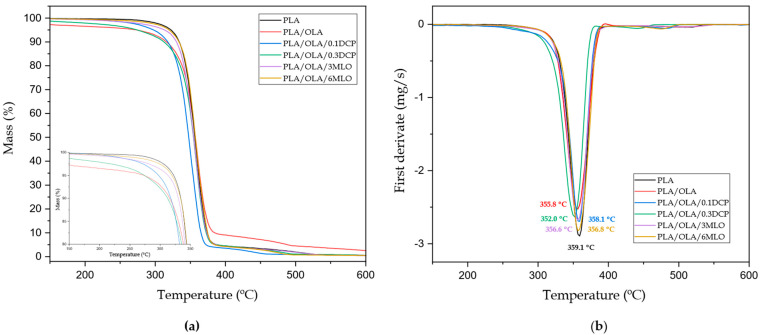
(**a**) Thermogravimetric analysis (TGA) curves and (**b**) first derivative (DTG) of neat PLA, base PLA/OLA blend and PLA/OLA blends subjected to reactive extrusion (REX).

**Figure 7 polymers-14-01874-f007:**
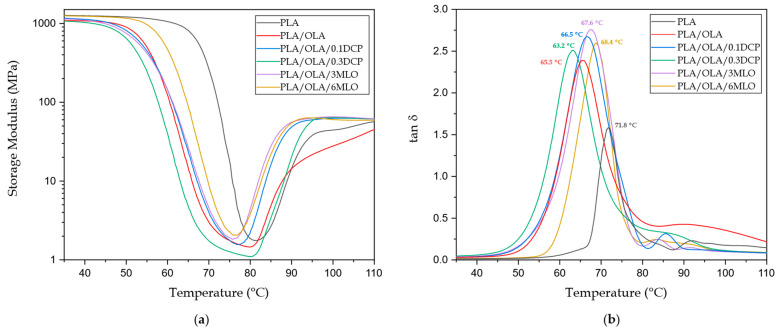
Plot evolution of (**a**) the storage modulus, G’ and (**b**) the dynamic damping factor (tan δ) of neat PLA, base PLA/OLA blend and PLA/OLA blends subjected to reactive extrusion (REX).

**Figure 8 polymers-14-01874-f008:**
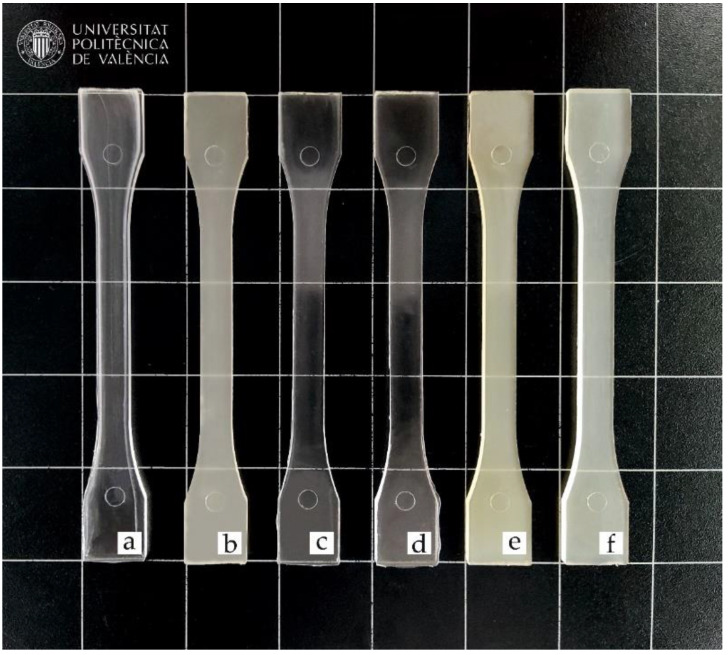
Visual appearance of the samples: (**a**) neat PLA; (**b**) PLA/OLA; (**c**) PLA/OLA/0.1DCP; (**d**) PLA/OLA/0.3DCP; (**e**) PLA/OLA/3MLO; (**f**) PLA/OLA/6MLO.

**Figure 9 polymers-14-01874-f009:**
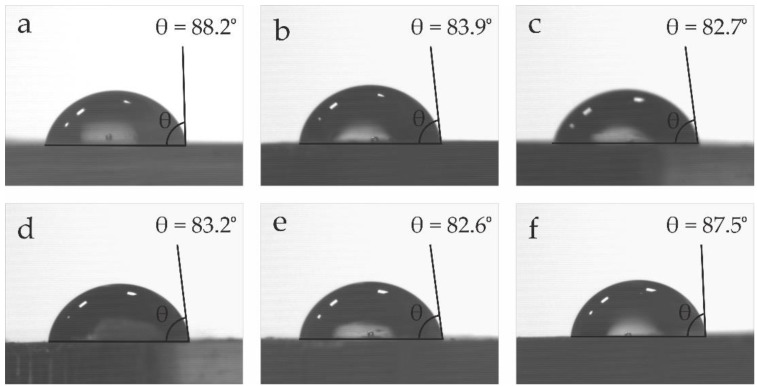
Water contact angle (θ_w_) of the samples: (**a**) neat PLA; (**b**) PLA/OLA; (**c**) PLA/OLA/0.1DCP; (**d**) PLA/OLA/0.3DCP; (**e**) PLA/OLA/3MLO; (**f**) PLA/OLA/6MLO.

**Table 1 polymers-14-01874-t001:** Summary of compositions according to the weight content (wt.%) of PLA/OLA and phr (parts by weight of additive per one hundred parts by weight of the base PLA/OLA formulation (90/10).

Code	PLA (wt. %)	OLA (wt. %)	DCP (phr)	MLO (phr)
PLA	100	0	0	0
PLA/OLA	90	10	0	0
PLA/OLA/0.1DCP	90	10	0.1	0
PLA/OLA/0.3DCP	90	10	0.3	0
PLA/OLA/3MLO	90	10	0	3
PLA/OLA/6MLO	90	10	0	6

**Table 2 polymers-14-01874-t002:** Summary of mechanical properties of the injection-moulded samples of neat PLA, base PLA/OLA blend and PLA/OLA blends subjected to reactive extrusion (REX). Tensile modulus (E), maximum tensile strength (σ_max_), elongation at break (ε_b_), Shore D hardness and impact strength (Charpy test).

Code	E (MPa)	σ_max_ (MPa)	ε_b_ (%)	Shore D Hardness	Impact Strength (kJ/m^2^)
PLA	2912 ± 84	47.0 ± 1.0	7.1 ± 0.3	81.6 ± 0.5	39.3 ± 3.3
PLA/OLA	3138 ± 45	30.8 ± 2.6	4.4 ± 0.4	77.8 ± 1.3	42.4 ± 2.4
PLA/OLA/0.1DCP	2996 ± 34	35.7 ± 0.7	5.0 ± 0.3	80.4 ± 1.1	44.5 ± 2.9
PLA/OLA/0.3DCP	3027 ± 30	36.8 ± 2.3	5.6 ± 1.5	76.0 ± 0.7	51.7 ± 2.4
PLA/OLA/3MLO	2987 ± 124	29.0 ± 1.3	4.3 ± 0.1	81.8 ± 1.3	52.3 ± 2.6
PLA/OLA/6MLO	3131 ± 44	41.7 ± 4.3	8.1 ± 0.8	79.2 ± 0.8	59.5 ± 1.2

**Table 3 polymers-14-01874-t003:** Glass transition temperature (T_g_), cold crystallization temperature (T_cc_), melting temperature (T_m_) and degree of crystallinity (χ_c_%) of neat PLA, base PLA/OLA blend and PLA/OLA blends subjected to reactive extrusion (REX).

Code	T_g_ (°C)	T_cc_ (°C)	∆H_cc_ (J/g)	T_m_ (°C)	∆H_m_ (J/g)	χ_c_%
PLA	63.3 ± 1.2	-	-	173.4 ± 1.8	19.4 ± 1.5	20.7 ± 0.3
PLA/OLA	49.8 ± 1.3	97.3 ± 2.1	27.6 ± 2.1	172.1 ± 1.4	41.2 ± 4.1	16.2 ± 0.2
PLA/OLA/0.1DCP	50.5 ± 1.1	107.1 ± 1.9	30.9 ± 1.3	171.9 ± 2.0	37.4 ± 3.2	7.8 ± 0.4
PLA/OLA/0.3DCP	49.8 ± 0.4	115.8 ± 2.3	39.2 ± 1.7	171.5 ± 2.1	41.2 ± 3.5	2.4 ± 0.2
PLA/OLA/3MLO	51.0 ± 1.5	97.6 ± 2.2	29.0 ± 1.1	172.0 ± 2.3	42.2 ± 2.9	15.8 ± 0.4
PLA/OLA/6MLO	56.5 ± 2.3	105.4 ± 2.8	34.2 ± 0.7	173.6 ± 2.1	39.8 ± 2.6	6.7 ± 0.2

**Table 4 polymers-14-01874-t004:** Main thermal degradation parameters of neat PLA, base PLA/OLA blend and PLA/OLA blends subjected to reactive extrusion (REX), in terms of the onset degradation temperature at a mass loss of 5 wt.% (T_5%_), maximum degradation rate peak temperature (T_deg_), and residual mass at 700 °C.

Code	T_5%_ (°C)	T_deg_ (°C)	Residual Mass (%)
PLA	321.6 ± 2.6	359.1 ± 2.1	0.10 ± 0.01
PLA/OLA	287.6 ± 3.3	355.8 ± 3.1	0.44 ± 0.02
PLA/OLA/0.1DCP	282.3 ± 2.2	358.1 ± 1.6	0.11 ± 0.01
PLA/OLA/0.3DCP	275.0 ± 1.4	352.0 ± 1.8	0.10 ± 0.01
PLA/OLA/3MLO	312.3 ± 2.8	356.6 ± 2.3	0.12 ± 0.01
PLA/OLA/6MLO	316.7 ± 1.2	356.8 ± 1.9	0.13 ± 0.01

**Table 5 polymers-14-01874-t005:** Dynamic mechanical properties of injection-moulded samples of neat PLA, base PLA/OLA blend and PLA/OLA blends subjected to reactive extrusion (REX), obtained from DMTA characterization at different temperatures.

Code	G’ (MPa) at 40 °C	G’ (MPa) at 80 °C	G’ (MPa) at 100 °C	T_g_ PLA (°C) *
PLA	1251 ± 35	1.6 ± 0.3	43.8 ± 1.5	71.8 ± 0.5
PLA/OLA	1076 ± 28	1.3 ± 0.2	27.8 ± 0.7	65.5 ± 0.8
PLA/OLA/0.1DCP	1129 ± 31	1.8 ± 0.2	62.9 ± 2.1	66.5 ± 0.9
PLA/OLA/0.3DCP	1032 ± 25	1.1 ± 0.3	63.8 ± 1.7	63.2 ± 0.6
PLA/OLA/3MLO	1102 ± 19	3.6 ± 0.1	59.7 ± 3.1	67.6 ± 0.7
PLA/OLA/6MLO	1235 ± 29	3.4 ± 0.3	65.7 ± 2.9	68.4 ± 0.7

* The T_g_ has been measured using the tan δ peak maximum criterion.

**Table 6 polymers-14-01874-t006:** Luminance and colour coordinates CIE-L*a*b*, of neat PLA, base PLA/OLA blend and PLA/OLA blends subjected to reactive extrusion (REX).

Code	L*	a*	b*	Yellowness Index (YI)
PLA	46.0 ± 0.0	−0.25 ± 0.01	1.92 ± 0.17	8.2 ± 0.3
PLA/OLA	43.7 ± 0.1	−0.74 ± 0.03	4.35 ± 0.08	17.6 ± 0.1
PLA/OLA/0.1DCP	44.7 ± 0.0	−0.17 ± 0.02	3.26 ± 0.04	10.9 ± 0.3
PLA/OLA/0.3DCP	45.5 ± 0.1	−0.08 ± 0.01	3.27 ± 0.04	11.1 ± 0.2
PLA/OLA/3MLO	40.1 ± 0.2	−1.61 ± 0.08	6.04 ± 0.18	23.2 ± 0.3
PLA/OLA/6MLO	41.4 ± 0.2	−0.96 ± 0.03	5.92 ± 0.16	20.7 ± 0.1

## Data Availability

Not applicable.

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
