# Peer review of "Development and Characterization of Polylactide Blends with Improved Toughness by Reactive Extrusion with Lactic Acid Oligomers"

_polymers, 2022, doi:10.3390/polym14091874_

Round 1

Reviewer 1 Report

In general, in my opinion, the manuscript is well prepared, and I have only а few recommendations for the authors.

The Abstract is somewhat extended. Try to shorten it, emphasizing only the study's relevance and the main results achieved.

At the end of the Introduction, it would be good to highlight the novelty of this study even more.

Figure 1 is of rather unsatisfactory quality, so please correct it. I want to ask if this figure is from the authors? If not, the reference from which it is taken should be given.

Please justify the parameters of the study more clearly. That is, to place them in the context of similar studies (to add references) and explain the selected values.

Please improve the quality (resolution) of Figure 2. The same applies to Figures 5, 6 and 7.

In the Conclusion part, it will be good if the authors emphasize the main novelty and contribution of the research more strongly.

Author Response

Reviewer 1

In general, in my opinion, the manuscript is well prepared, and I have only а few recommendations for the authors.

  1. The Abstract is somewhat extended. Try to shorten it, emphasizing only the study's relevance and the main results achieved.

ANSWER

Thank you for the appreciation. In order to improve the quality of the paper, the abstract has been reduced.

  1. At the end of the Introduction, it would be good to highlight the novelty of this study even more.

ANSWER

Following the recommendations of the reviewer, new information has been added in the introduction to further highlight the novelty of this study.

  1. Figure 1 is of rather unsatisfactory quality, so please correct it. I want to ask if this figure is from the authors? If not, the reference from which it is taken should be given.

ANSWER

Thank you for the recommendation. We have replaced the low-resolution Figure 1 with a new higher resolution image. In answer to your question, yes, the figure has been made by one of the authors

  1. Please justify the parameters of the study more clearly. That is, to place them in the context of similar studies (to add references) and explain the selected values.

ANSWER

The parameters of the article have been selected based on the following previous works.

Lascano, D.; Moraga, G.; Ivorra-Martinez, J.; Rojas-Lema, S.; Torres-Giner, S.; Balart, R.; Boronat, T.; Quiles-Carrillo, L. Development of injection-molded polylactide pieces with high toughness by the addition of lactic acid oligomer and characterization of their shape memory behavior. Polymers 2019, 11, 2099.

Rojas-Lema, S.; Quiles-Carrillo, L.; Garcia-Garcia, D.; Melendez-Rodriguez, B.; Balart, R.; Torres-Giner, S. Tailoring the properties of thermo-compressed polylactide films for food packaging applications by individual and combined additions of lactic acid oligomer and halloysite nanotubes. Molecules 2020, 25, 1976.

As the reviewer has commented, in order to improve the quality of the article, some references have been added in the parameters section to justify the decision.

  1. Please improve the quality (resolution) of Figure 2. The same applies to Figures 5, 6 and 7.

ANSWER

Thanks again for your appreciation, as well as Figure 1, Figures 2, 5 and 6 have been replaced by new ones with a higher resolution.

  1. In the Conclusion part, it will be good if the authors emphasize the main novelty and contribution of the research more strongly.

ANSWER

Your advice has been considered and changes have been made to highlight the contribution of this work and its novelty.

Reviewer 2 Report

Polymers

In their research article titled “Development and characterization of polylactide blends with improved toughness by reactive extrusion with lactic acid
Oligomers” The submitted study comprehensively analyzes PLA-based formulations with improved toughness by blending with lactic acid oligomers. The literature review and the importance of the lactic acid oligomers explain systematically; however, some paragraphs of the introduction are very lengthy. I encourage their efforts; however, modify your manuscript according to the following changes. I think it could be published with minor revisions.

Q 1. Graph-1 needs to draw again.

Q 2. REX with DCP increased both modulus and tensile strength, but the elongation at break of the ternary blend was lower than that of neat PLA; why?

Q 3. The quality of figures 2, 3, and 5 are not good, and please redraw them.

Q 4. As shown in Figure 6a, the thermal stability of the PLA/OLA blends is reduced due to the lower molecular weight of OLA molecules compared to PLA polymer chains. Please make the relation between molecular weight and  thermal properties

Author Response

In their research article titled “Development and characterization of polylactide blends with improved toughness by reactive extrusion with lactic acid Oligomers” The submitted study comprehensively analyzes PLA-based formulations with improved toughness by blending with lactic acid oligomers. The literature review and the importance of the lactic acid oligomers explain systematically.

  1. however, some paragraphs of the introduction are very lengthy. I encourage their efforts; however, modify your manuscript according to the following changes. I think it could be published with minor revisions.

ANSWER

Thank you for the recommendation. Following your advice, some of the longest paragraphs of the introduction have been shortened

  1. Q 1. Graph-1 needs to draw again.

ANSWER

As indicated by the reviewer, we have improved the resolution of graph 1

Q 2. REX with DCP increased both modulus and tensile strength, but the elongation at break of the ternary blend was lower than that of neat PLA; why?

  1. ANSWER

According to the reviewer’s suggestion, explanation has been added to the above question.

  1. Q 3. The quality of figures 2, 3, and 5 are not good, and please redraw them.

ANSWER

Following the recommendations of the reviewer, the quality and size of the figures have been modified to improve the clarity and quality of the work.

  1. Q 4. As shown in Figure 6a, the thermal stability of the PLA/OLA blends is reduced due to the lower molecular weight of OLA molecules compared to PLA polymer chains. Please make the relation between molecular weight and thermal properties

ANSWER

Your advice has been considered and changes have been made in order to explain the relationship between molecular weight and thermal degradation.

Reviewer 3 Report

The about development and characterization of polylactide blends with improved toughness by  reactive extrusion with lactic acid oligomers is well written and can be of interest to the readers of polymers. A few issues need to be addressed before it can be published. More in details

Line 103-105 there are a few typos please check

Figure 1 the resolution is quite low and the labels are difficult to read

Color and wetting characterization should be separate (section 2.3.5)

Table 3 as a note even if it is usual to represent the numbers with two significant digits an value of 39.2 +- 1.7 should be correctly represented as 39+-2. Using scientific notation will avoid all the problems The same apply also to the other tables

Figure 8. As a note the authors can increase the visual info given to the reader including a calibration chart in the photo (see Gretag or other specific calibration standard)

Author Response

Reviewer 3

The about development and characterization of polylactide blends with improved toughness by reactive extrusion with lactic acid oligomers is well written and can be of interest to the readers of polymers. A few issues need to be addressed before it can be published. More in details

  1. Line 103-105 there are a few typos please check

ANSWER

Thank you for your comments, we have corrected the various typos in the lines mentioned above, and we have also reviewed the literature of the work.

  1. Figure 1 the resolution is quite low and the labels are difficult to read

ANSWER

As indicated by the reviewer, we have improved the resolution of Figure 1

  1. Color and wetting characterization should be separate (section 2.3.5)

ANSWER

Your advice has been considered and changes have been made in order to separate color and wetting properties

  1. Table 3 as a note even if it is usual to represent the numbers with two significant digits and value of 39.2 +- 1.7 should be correctly represented as 39+-2. Using scientific notation will avoid all the problems The same apply also to the other tables

ANSWER

thank you for your observation, we have corrected the digits of the different tables of the work, according to the number of representative decimals for each value.

  1. Figure 8. As a note the authors can increase the visual info given to the reader including a calibration chart in the photo (see Gretag or other specific calibration standard)

ANSWER

The purpose of the figure is to highlight the differences in opacity between the different samples. We have not thought it necessary to include a calibration chart in the figure. Accurate colour information using CIE-L*a*b* coordinates is shown in Table 6.